# Lost in Context: Addressing Context Anxiety in Large Language Models

**Ifueko Igbinedion** [1]  **Jillian Ross** [1]  **Etienne Ricardez** [1]  **Sertac Karaman** [1]  **Eric So** [1]

## Abstract

Conventional wisdom suggests that reasoning models fail when problems exceed their capabilities. However, we find that frontier reasoning models sometimes possess the necessary capabilities to solve problems but fail due to premature self-doubt – a phenomenon informally known as context anxiety. We provide the first systematic study of context anxiety, demonstrating that it arises, in part, from a model's inability to accurately estimate the tokens required to complete a task. We also show that context anxiety leads to material efficiency losses when models operate under perceived constraints. Building on this analysis, we further show that models can learn alternative strategies for solving long-horizon problems without exhibiting context anxiety, suggesting that performance improvements may be achievable not through scaling model capabilities, but by improving models' ability to accurately assess and adapt to their own limitations.

## 1. Introduction

The rapid advancement of large language models has led to impressive gains in multi-step reasoning capabilities, with frontier models now capable of solving problems that require extended chains of thought and long-horizon planning. Advances in training and inference, including policy optimization over sampled reasoning traces and inference-time aggregation of candidate rationales, have further improved performance in difficult domains such as mathematics and programming (Guo et al., 2025; Wang et al., 2023c). Yet a puzzling pattern has emerged: models sometimes fail not because they lack the capabilities to solve a problem, but

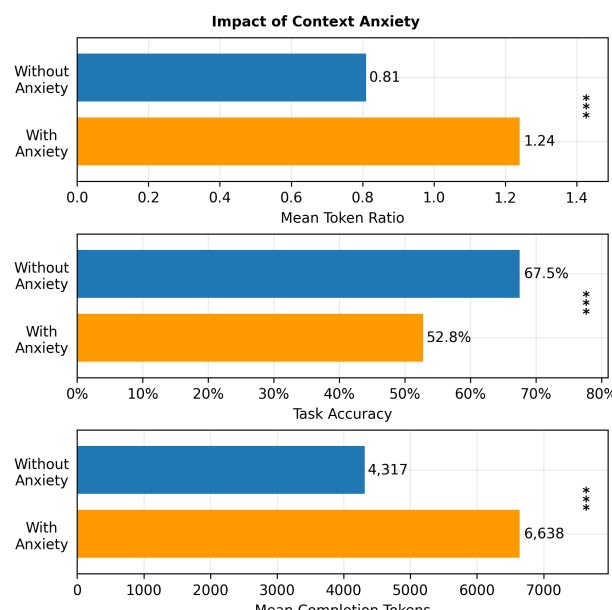

*(a)* Models with context anxiety overestimate token requirements (top), leading to reduced task accuracy overall (middle) and reduced token efficiency when solutions succeed (bottom).

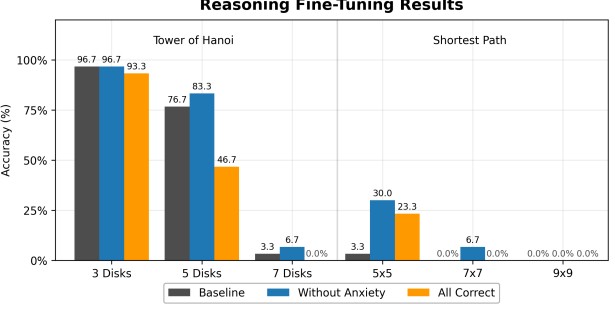

*(b)* Fine-tuning on reasoning traces exhibiting no anxiety improves task accuracy compared to fine tuning on all correct samples.

*Figure 1.* Context anxiety stems from miscalibrated token estimation and creates a reduction in performance and efficiency (a), but context anxiety can be reduced through supervised fine tuning on reasoning traces without anxiety (b).

because they believe they do. This raises a critical question: do models fail because problems are genuinely too complex, or because they believe they are?

When faced with challenging problems, models sometimes

[1]Massachusetts Institute of Technology, Cambridge, MA, USA. Correspondence to: Ifueko Igbinedion <ifueko@mit.edu>, Jillian Ross <jillianr@mit.edu>, Etienne Ricardez <etienne7@mit.edu>, Sertac Karaman <sertac@mit.edu>, Eric So <eso@mit.edu>.

abandon tasks with worries that the solution to the task will exceed the tokens they have available. This same worry seems to reduce efficiency: models produce solutions less concisely. We describe this phenomenon as *context anxiety* – when models worry about their ability to solve the task within their token limits, despite having sufficient tokens to complete them. Though this worry can manifest in different ways, underlying both manifestations is a model's inability to accurately estimate the tokens required to complete a task.

Understanding the distinction between failures due to context anxiety and failures due to genuine inability has practical implications for improving model performance. If failures stem from context anxiety rather than insufficient capabilities, then performance gains may be achievable not through scaling model size or training compute, but through improving models' ability to accurately assess their own limitations.

This paper introduces the first systematic methodology for detecting and measuring context anxiety in large language models. We develop protocols to distinguish **anxiety-driven failures** (premature abandonment with claims of resource insufficiency) from **capability-driven failures** (genuine inability despite attempted solutions). We establish that a model's **miscalibration of estimated token usage predicts the manifestation of context anxiety**. Applying this methodology to the Tower of Hanoi puzzle, we find that frontier reasoning models expressing context anxiety misestimate their token usage by 24%, and this miscalibration predicts 15% lower accuracy and drives a 54% increase in token usage when solutions succeed. Importantly, we observe this behavior **even in models designed to reason over long contexts** and adapt their generation length dynamically, including recent context-aware models (Anthropic, 2025). Finally, leveraging our anxiety measurement method, we show that lightweight supervised fine tuning over reasoning traces reduces context anxiety by over 50%, demonstrating that this failure mode is **behaviorally mutable rather than capability limited**. These findings suggest that accurate self-assessment, not just raw reasoning capabilities, fundamentally shapes reasoning model performance under extended contexts. We further validate these findings on a second long-horizon task—shortest-path grid search—observing qualitatively similar patterns (Appendix D).

## 2. Related Work

Context anxiety emerges at the intersection of reasoning capability and self-assessment. This section reviews work on multi-step reasoning methods, model behavior under contextual stress, and emergent patterns that shape when models engage with or abandon challenging tasks.

### 2.1. From Chain-of-Thought Prompting to Robust Multi-Step Reasoning

Large language models have progressed from generating coherent text to addressing tasks requiring multi-step reasoning. While scaling and instruction tuning have expanded what models can solve, performance depends not only on underlying capability but also on whether models reliably engage in extended problem solving when necessary.

Chain-of-thought (CoT) prompting shows that instruction-tuned LLMs can substantially improve performance on multi-step tasks—without additional fine-tuning—by generating intermediate rationales during inference (Wei et al., 2022b; Kojima et al., 2022). Subsequent work studies which properties of CoT matter most and how example selection and formatting further amplify these gains (Fu et al., 2023).

However, CoT does not guarantee faithful reasoning: models can produce fluent chains that increase surface plausibility while remaining weakly coupled to relevant evidence or the final decision, particularly under distribution shift or increased task difficulty (Wang et al., 2023a). This motivates approaches that impose structure or provide additional support during reasoning. For knowledge-intensive tasks, retrieval can be interleaved with reasoning so that intermediate claims are supported by evidence (Trivedi et al., 2023). Task decomposition strategies such as least-to-most prompting improve generalization by solving simpler subproblems before attempting the full task (Zhou et al., 2023). Reasoning traces can also be used as supervision, with larger models acting as teachers whose rationales are distilled into smaller models (Ho et al., 2023).

Beyond supervision, alignment fine-tuning leverages feedback over multiple reasoning traces, encouraging models to prefer successful rationales and correcting miscalibration in internal scoring (Wang et al., 2023b). At inference time, robustness can be improved by aggregating across diverse reasoning paths, such as self-consistency and step-aware verification (Wang et al., 2023c; Li et al., 2023). At the same time, longer reasoning increases latency and compute, motivating work on controlling verbosity and reducing redundant overthinking (Sui et al., 2025). Together, these results show that eliciting reasoning alone is insufficient: failures in calibration, grounding, and effort allocation persist even when models possess sufficient task-relevant capability.

### 2.2. Eliciting Reliable Reasoning under Contextual Stress

Even with advances in prompting, supervision, and decoding, substantial evidence suggests a mismatch between surface fluency and underlying reliability across regimes of difficulty. On harder problems or in longer contexts, models often maintain coherent intermediate steps while answer ac-

curacy collapses (Levy et al., 2024; Liu et al., 2024). Moreover, the same model can produce responses with widely varying lengths and degrees of elaboration, indicating instability in how much reasoning is allocated to similar inputs (Zheng et al., 2023). These failures reflect not only missing information, but shifts in behavior under contextual or cognitive stress and sensitivity to superficial prompt variations (Errica et al., 2025).

Work on calibration and self-knowledge further shows that models exhibit both sampling-related and competence-related uncertainty. While models can sometimes estimate whether they are likely to answer correctly, these self-assessments are task-dependent and often fail to generalize across formats or domains (Kadavath et al., 2022; Yin et al., 2023; Kapoor et al., 2024; Zhou et al., 2024). As a result, models may display overconfidence on unfamiliar tasks or reduce effort prematurely when tasks appear difficult, even when sufficient information remains available.

Long-context evaluations provide a clear lens into these effects. When task content is fixed but context length or structure varies, models exhibit substantial degradation and systematic changes in response behavior (Levy et al., 2024; Liu et al., 2024; Li et al., 2024). These changes often appear as qualitative mode switches, including excessive elaboration, omission of critical steps, or premature termination (Levy et al., 2024; Sui et al., 2025; Su et al., 2025; Yang et al., 2025). Generated reasoning traces may also be weakly coupled to internal decision processes, creating an illusion of deliberation even when answers are incorrect (Turpin et al., 2023). Notably, related behaviors have also been observed in recent models explicitly designed for long-context reasoning, indicating that such failures are not restricted to models with limited context windows but can also arise in general purpose foundation models without explicit context management mechanisms (Marcu & The Cognition Team).

Taken together, these results suggest that reasoning performance is limited not only by representational or computational capacity, but by how models estimate task difficulty, solvability, and required effort under uncertainty. We view these interacting failures—miscalibration, unstable effort allocation, and context-sensitive mode switching—as manifestations of a broader behavioral phenomenon that can be described as context anxiety, in which perceived constraints alter how models engage in extended reasoning. This framing motivates our focus on token and effort estimation as mechanistic contributors to when models engage, overthink, or prematurely disengage in long-context settings.

### 2.3. Emergent Behavioral Patterns under Context Anxiety

Prior work has documented emergent abilities in large language models, where qualitatively new behaviors appear as scale increases despite not being explicitly trained (Wei et al., 2022a; Berti et al., 2025). There is active debate over whether such behaviors reflect genuine phase transitions or instead result from evaluation thresholds or sensitivity to task formulation (Schaeffer et al., 2023; Lu et al., 2024). Regardless of cause, these findings show that model behavior can change abruptly across conditions.

Emergent behaviors have been observed in reasoning-like skills such as analogy, where performance improves sharply beyond certain model scales(Webb et al., 2023), suggesting that apparent gains may reflect shifts in behavioral strategies or internal control policies. Scaling and stronger supervision can also amplify undesirable behaviors: models may overthink simple problems or disengage under uncertainty (Sui et al., 2025), and beliefs about competence influence whether models attempt, defer, or hedge (Kadavath et al., 2022; Yin et al., 2023; Kapoor et al., 2024).

Long-context settings make these shifts especially visible. When context length or structure changes while task content remains fixed, models exhibit systematic performance drops and behavioral changes (Levy et al., 2024; Liu et al., 2024; Li et al., 2024). Training objectives can further shape behavioral preferences: alignment and preference-based fine-tuning may encourage rationales that appear plausible even when they are not correct (Wang et al., 2023b). We hypothesize that many of these behavioral shifts arise at inference time under contextual stress, driven by mismatches between perceived and actual constraints–particularly in estimates of required effort and available token budgets. These failures can be understood as manifestations of context anxiety.

To investigate this mechanism in a controlled setting, we use the Tower of Hanoi task to isolate long-horizon planning under varying perceived constraints, enabling direct measurement of engagement, effort allocation, and associated failure modes.

## 3. Method

We introduce a systematic approach to quantify and mitigate context anxiety in large language models. Our methodology consists of three components: (1) automatic detection of context anxiety related reasoning behavior, (2) measurement of token usage calibration, and (3) a lightweight reasoning fine-tuning protocol for context anxiety mitigation without training new behaviors.

### 3.1. Detection Protocol

To identify context anxiety, we analyze model outputs for explicit statements indicating the task is "too much" to process in their token limits. To reduce sensitivity to specific phrasing, we employ a panel of LLMs to classify reasoning traces for the presence of semantic equivalents for phrases like

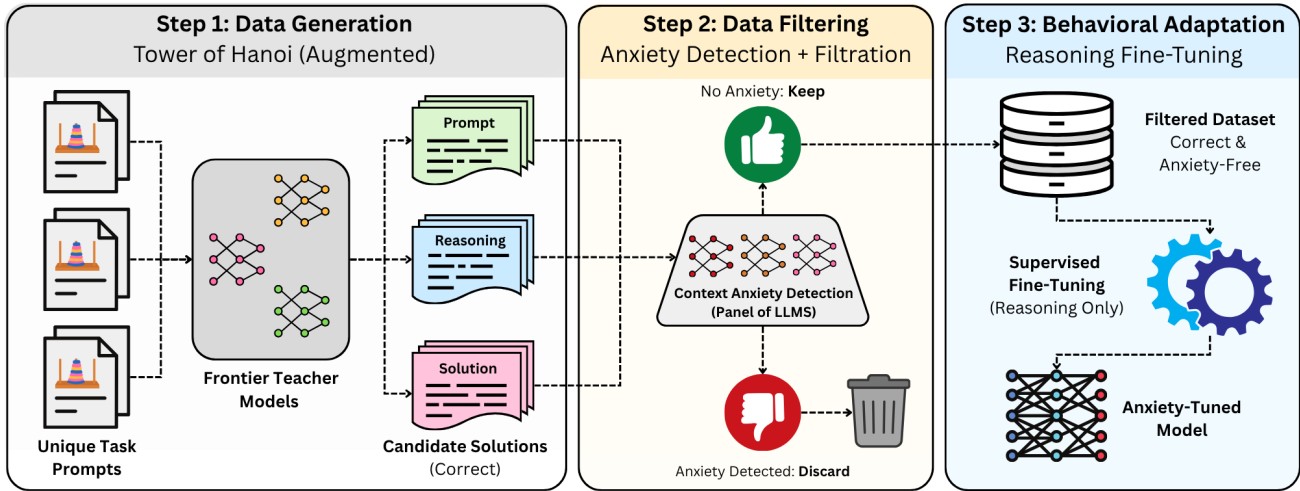

*Figure 2.* **Mitigating context anxiety via selective reasoning fine-tuning.** Our pipeline consists of three steps. **Step 1 (Data Generation):** Frontier teacher models generate candidate solutions—comprising prompts, reasoning traces, and final solutions—for augmented Tower of Hanoi tasks. **Step 2 (Data Filtering):** A panel of LLMs evaluates the generated reasoning to detect context anxiety. Responses exhibiting anxiety are discarded, while correct, anxiety-free reasoning traces are retained. **Step 3 (Behavioral Adaptation):** We perform Supervised Fine-Tuning (SFT) exclusively on the filtered dataset of reasoning traces to produce the final anxiety-tuned model.

"exceeds my capacity," "too complex to solve completely," or "beyond my context window." We use the following protocol:

$$C = \left\lceil \frac{1}{J} \sum_{k=1}^{J} J_k(r) \right\rceil \qquad (1)$$

where $J_k(r)$ is the strength of context anxiety that judge model $J_k$ detects in reasoning trace $r$. This automated classification allows us to systematically distinguish between:

- **Anxiety-driven failures**: The model abandons the task while claiming resource insufficiency

- **Capability-driven failures**: The model attempts the task but produces incorrect solutions

- **Successful attempts**: The model completes the task with or without expressing difficulty

This separation enables downstream analyses that isolate behavioral disengagement from problem-solving failures. Sensitivity and human-validation analyses of the detector are reported in Appendix F.

### 3.2. Calibration Measurement

To test whether context anxiety stems from miscalibration of expected token use, we measure how accurately models estimate their token requirements. We introduce the *winsorized token ratio* $T$:

$$T = \frac{\text{Estimated tokens required}}{\text{Actual tokens used}} \qquad (2)$$

where estimated tokens required is the number of tokens the model thinks it has used to generate its output. A ratio of 1.0 indicates perfect calibration; values greater than 1 indicate overestimation; values less than 1 indicate underestimation. We use winsorization at the 1th and 99th percentiles to reduce the influence of outliers.

Estimated token usage is elicited by prompting the same model to provide a numeric estimation of how many tokens it believes were required to complete the solution after generation is finished. While models do not have access to tokenizer internals, this estimate reflects perceived generation effort rather than true computational cost, which is precisely the quantity that should influence behavioral disengagement. Actual token counts are computed from the model's respective tokenizer. This evaluation allows us to quantify systematic biases in perceived effort relative to true generation cost.

We analyze the correlation between overestimation of required tokens and context anxiety and whether miscalibration mediates the relationship between anxiety and task accuracy.

### 3.3. Behavioral Adaptation for Long-Context Anxiety Reduction

To test whether context anxiety reflects a modifiable behavioral policy rather than a fixed capacity limitation, we evaluate a lightweight fine-tuning approach designed to

reduce anxiety-driven disengagement without increasing model size or inference-time compute.

We construct a distillation dataset from multiple teacher models, including both the target model and stronger frontier models. For each prompt, we filter responses to include only those that successfully complete the task and exhibit no detected context anxiety under the protocol described above. Each training example consists of the original prompt and the full reasoning trace. We then fine-tune the student model using standard supervised fine tuning on the filtered dataset, optimizing reasoning strategies without directly optimizing solutions. Figure 2 summarizes this process.

This procedure encourages the model to adopt long-horizon strategies that solve difficult instances without invoking perceived resource constraints. This protocol isolates behavioral adaptation from capability expansion: the model is not given any new information or prompted with knowledge of a longer context, but is instead trained to follow non-anxious solution strategies when faced with identical task distributions. We evaluate whether this fine-tuned model exhibits reduced rates of context anxiety, improved calibration of token estimates, and improve accuracy-efficiency tradeoffs under long-context stress.

## 4. Results

We apply this methodology to the Tower of Hanoi puzzle, a well-suited domain for studying context anxiety because:

1. **Systematic scaling**: Difficulty increases exponentially with the number of disks ($n$), requiring exactly $2^n - 1$ moves for optimal solutions

2. **Known resource requirements**: The solution space is deterministic, allowing us to verify whether tasks truly exceed model capabilities

3. **Sequential reasoning demands**: The puzzle requires multi-step planning that exercises extended reasoning chains

We test 30 unique problems per disk, ranging from 2 to 12 disks. Our evaluation set has 270 problems that span from trivial (7 moves) to challenging (4,095 moves).

We evaluate a variety of open-source and closed-source models: DeepSeek R1 (Guo et al., 2025), Claude Sonnet 3.7, Claude Sonnet 4.5, Gemini 2.5 Flash (Comanici et al., 2025), Kimi K2 Thinking, and o4 Mini. All problems in our test set remain well within the maximum token limits of modern frontier models (typically 60K–128K tokens). This yields a total of 1,620 observations, of which 1,585 were available for analysis due to API errors.

| Model | % Anxiety | % Capability |
|---|---|---|
| Claude Sonnet 3.7 | **73.4%** | 26.6% |
| Claude Sonnet 4.5 | 10.7% | **89.3%** |
| DeepSeek R1 | **66.3%** | 33.7% |
| Gemini 2.5 Flash | 5.1% | **94.9%** |
| Kimi K2 Thinking | 41.6% | **58.4%** |
| o4 Mini | 19.7% | **80.3%** |

*Table 1.* Percentage of failures that are anxiety-driven vs. capability-driven.

### 4.1. LLMs have context anxiety

Frontier models all demonstrate some form of context anxiety. Further, context anxiety tends to increase with task complexity, as shown in Figure 3. Some models, i.e. Claude Sonnet 3.7 and DeepSeek R1, refuse nearly 100% of the most difficult problems in our dataset. The same qualitative pattern is observed on a second task family, shortest-path grid search (Appendix D).

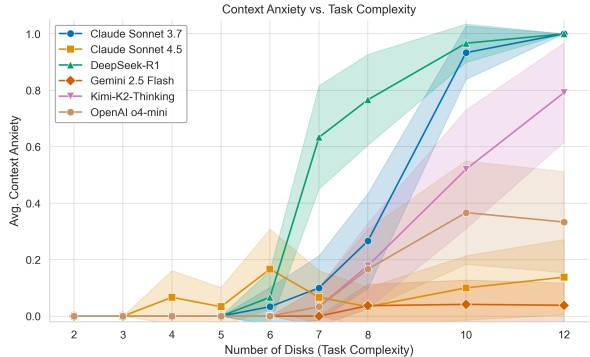

*Figure 3.* As task complexity increases, frontier models generally struggle to successfully complete the Tower of Hanoi.

To distinguish between failures due to disengagement and failures due to genuine problem solving challenges, we further classify unsuccessful attempts into anxiety-driven and capability driven failures. This breakdown allows us to verify that many failures at higher disk counts are attributable to explicit disengagement rather than unsuccessful attempts at completing the task. As seen in Table 1, we see that the majority of failures of DeepSeek R1 and Claude Sonnet 3.7 are anxiety-driven, while the majority of failures of Claude Sonnet 4.5, Gemini 2.5 Flash, Kimi K2 Thinking, and o4 Mini are capability-driven.

### 4.2. Overestimating problem demands leads to context anxiety

What causes models to give up on problems they could theoretically solve? We hypothesize that context anxiety stems from poor calibration: models with context anxiety

systematically overestimate how many tokens a problem will require, leading them to preemptively declare tasks unsolvable. To test this, we measure how accurately models estimate their token usage through the *winsorized token ratio*, as described in Section 3.2.

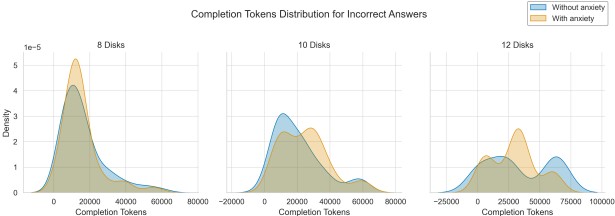

*(a)* When models generate incorrect solutions, those with context anxiety consume more tokens than those without.

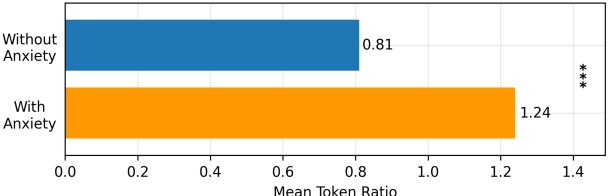

*(b)* Models overestimate required tokens with context anxiety but underestimate without it. A ratio of 1 indicates perfect calibration.

*Figure 4.* Impact of context anxiety on token usage.

The results strongly support our hypothesis. As shown in Figure 4b, models that exhibit context anxiety overestimate their token requirements by approximately 24%. In contrast, models that attempt problems without expressing context anxiety underestimate their token requirements by 19%. The difference in estimation error between models with and without context anxiety is significant ($p < 0.01$), indicating that overestimation of token requirements is significantly higher in the presence of context anxiety. [1]

We hypothesize that this miscalibration acts as a self-fulfilling prophecy: models predict a task will be difficult, overestimate the token requirements required to complete the task, and prematurely terminate before any genuine attempt is made. The problem is not that tasks are too demanding— it is that models perceive them to be. Conversely, when models predict a task will be easier, they underestimate token requirements, and this optimism leads them to actually attempt the problem.

### 4.3. Context anxiety decreases accuracy and token efficiency

The consequences of context anxiety extended beyond reduced accuracy. When models worry they lack sufficient resources and abandon problems prematurely, they fail to pro-

duce correct solutions, even for problems well within their capabilities. However, worry about resource constraints also harms token efficiency, producing correct answers less concisely and with longer response times.

To quantify the impact of context anxiety on task accuracy and efficiency while controlling for baseline model capabilities and inherent task difficulty, we run an econometric analysis with task complexity and model fixed effects:

$$Y_{idm} = \alpha + \beta_1 C_{idm} + \mu_m + \gamma_d + \varepsilon_{idm} \qquad (3)$$

where $Y_{idm}$ denotes the accuracy for problem $i$ with $d$ disks solved by model $m$. We construct the context anxiety indicator $C_{idm}$ through semantic analysis as detailed in Section 3.1. The intercept $\alpha$ captures baseline accuracy, while $\beta_1$ quantifies the marginal effect of context anxiety on performance.

The inclusion of model fixed effects $\mu_m$ and disk fixed effects $\gamma_d$ controls for heterogeneity in model capabilities and varying task complexity. By including both sets of fixed effects, we ensure that $\beta_1$ identifies the impact of context anxiety on accuracy, purged of confounding variation from model capability and problem complexity.

Table 2 presents our regression analysis separating the effect of context anxiety from two potential confounds. The results show that context anxiety reduces accuracy by 15.3% ($p < 0.01$), even after accounting for task complexity and model-specific performance differences. Per-model, per-difficulty accuracy is reported in Appendix G.

Context anxiety is also associated with material efficiency losses among successful solutions. Awareness of token constraints appears to limit the ability of models to generate efficient solution strategies. Using the econometric regression with model and disk fixed effects, we find that correct answers produced with context anxiety are 54%[2] longer ($p < 0.01$) compared to correct answers produced without context anxiety.

This dual effect suggests that context anxiety operates hurts performance on multiple fronts: it imposes pressure that reduces the likelihood of success, but when success is achieved, that same pressure yields less streamlined solutions.

### 4.4. Behavioral adaptation reduces context anxiety

Finally, we test whether context anxiety reflects a modifiable behavioral policy rather than a fixed limitation of model capacity. Using behavioral adaptation described in Section

---

[1]Per-model maximum output lengths used in evaluation are listed in Appendix E.

[2]We use the value of the intercept of the regression. In this case, the intercept refers to average value of completion tokens for when context anxiety is absent.

| | (1) Accuracy ($Y_{idm}$) | (2) Completion Tokens |
|---|---|---|
| Context Anxiety ($C_{idm}$) | $-0.153^{***}$ (0.026) | $2323.5^{***}$ (525.8) |
| Adj. $R^2$ | 0.676 | 0.666 |
| N | 1,585 | 1,007 |
| Model FE ($\mu_m$) | Yes | Yes |
| Disk FE ($\gamma_d$) | Yes | Yes |

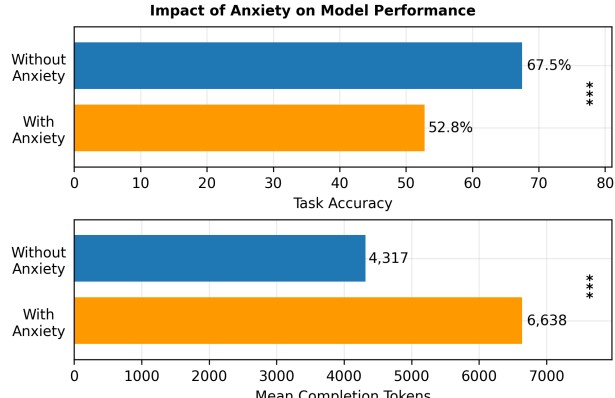

*Table 2.* The performance impact of context anxiety. By interpreting the coefficient in the fixed effects regression, we find that context anxiety reduces accuracy by 15.3% while increasing token usage by 2,323 tokens for correct answers. $^{***}$ indicates $p < 0.01$.

3.3, we fine-tune GPT-OSS-20B (OpenAI et al., 2025) on filtered examples from the same Tower of Hanoi dataset, retaining only generations that successfully complete the task without expressing context anxiety.

We perform supervised fine-tuning with loss restricted to the reasoning trace of each response, masking the final answer tokens from the training objective. This targets adaptation of the model's reasoning behavior rather than imitation of specific solution outputs. See Appendix C for specific training and hyperparameter details.

After fine tuning, the adapted model exhibits substantially less context anxiety compared to the baseline model. As shown in Figure 5b, the adapted model shifts toward reaching genuine capability limits, frequently producing exhaustive reasoning traces rather than prematurely abandoning the task. To isolate the factors driving this behavioral adaptation and evaluate the robustness of the learned policy, we perform a number of ablations. Table 3 summarizes the overall accuracy of the baseline, our anxiety-filtered Reasoning SFT, and an all-correct control across various held-out difficulties and domains. Overall, the Reasoning SFT model consistently matches or outperforms the baseline and the control, with the most pronounced gains observed on mid-tier difficulties and out-of-domain transfer tasks.

### 4.4.1. EVEN / ODD DISK HELD-OUT SPLIT

To test whether adaptation of reasoning policy transfers across difficulties rather than being specialized to the distilled disk counts, we fine-tune on filtered traces generated at even disk counts only ($n \in \{2, 4, 6, 8, 10, 12\}$) and evaluate on odd disk counts $n \in \{3, 5, 7\}$ that were never seen during fine-tuning. As shown in Table 3, the fine-tuned model improves or maintains accuracy on the held-out odd-disk evaluation set. Furthermore, Table 4 demonstrates that the detected anxiety rates decrease substantially across all held-

out difficulties. The direction of both the accuracy change and the anxiety reduction is preserved, consistent with the interpretation that fine-tuning modifies a disk-count-agnostic reasoning policy rather than memorized trajectories.

### 4.5. Cross-Task Transfer (Hanoi → Shortest Path)

We further evaluate the same Tower-of-Hanoi-distilled SFT checkpoint, without any shortest-path data in the fine-tuning mixture, on held-out shortest-path instances. Referring back to Table 3, the fine-tuned model retains a non-trivial accuracy improvement on the unseen task family (most notably a +26.7% absolute gain on 5×5 grids), indicating that the effect of fine-tuning is not confined to move-sequence imitation specific to Tower of Hanoi.

### 4.6. All-Correct SFT Control (No Anxiety Filtering)

Finally, to isolate the contribution of the anxiety-based filter used to construct the training set, we train an otherwise identical SFT checkpoint on *all* correct traces (without removing examples in which the reasoning trace exhibited detected context anxiety) using the same hyperparameters, data volume, and masking scheme. As shown in Table 3 and Figure 5c, the anxiety-filtered checkpoint severely outperforms the all-correct checkpoint on accuracy, reducing context anxiety and meeting or exceeding performance at all difficulty levels. These results indicate that the gains attributable to SFT are not fully explained by fine-tuning on successful traces in general, but rely specifically on the targeted behavioral filtering.

## 5. Conclusion

In this work, we present an empirical study of context anxiety as a measurable and systematic behavioral failure mode in large language models. While prior work has examined

| Task | Difficulty | Baseline | Reasoning SFT | | All-Correct SFT | |
|---|---|---|---|---|---|---|
| | | | Acc. | Δ | Acc. | Δ |
| Tower of Hanoi | 3 disks | *96.7%* | *96.7%* | +0.0 | 93.3% | −3.4 |
| | 5 disks | 76.7% | **83.3%** | +6.6 | 46.7% | −30.0 |
| | 7 disks | 3.3% | **6.7%** | +3.4 | 0.0% | −3.3 |
| Shortest Path | 5×5 | 3.3% | **30.0%** | +26.7 | 23.3% | +20.0 |
| | 7×7 | 0.0% | **6.7%** | +6.7 | 0.0% | +0.0 |
| | 9×9 | 0.0% | 0.0% | +0.0 | 0.0% | +0.0 |

*Table 3.* Accuracy across held-out tasks and difficulties. Δ denotes the absolute change relative to the baseline. Bold indicates the best-performing model for each task difficulty, and italics indicate non-zero ties.

*Table 4.* Detected anxiety rates (Anx.) for the Even-disks SFT evaluated on held-out odd disk counts.

| Model | 3 disks | 5 disks | 7 disks |
|---|---|---|---|
| Baseline | 0.05 | 0.28 | 0.63 |
| Reasoning SFT | 0.02 | 0.12 | 0.34 |

long-context brittleness and reasoning degradation under stress, our results suggest that a subset of these failures reflect an emergent behavioral response to perceived constraints, which we characterize as context anxiety and show can be measured and mitigated.

Using Tower of Hanoi as a controlled long-horizon planning task, we demonstrate that frontier reasoning models frequently abandon solvable problems while citing resource limitations, despite operating well within their available context windows. We further show that these failures are strongly associated with miscalibration in model estimates of required reasoning effort–particularly overestimation of token usage–and that this miscalibration predicts premature disengagement, reduced accuracy, and increased token usage.

Beyond diagnosis, we show that context anxiety is not a fixed property of reasoning models, but a behavioral regime that can be shifted. Lightweight supervised fine-tuning on non-anxious reasoning traces reduces premature task abandonment while preserving or improving both performance and efficiency, suggesting that models can learn alternative engagement strategies under perceived resource constraints. This points to a complementary axis of improvement to scaling and stronger reasoning supervision: improving how models assess and respond to their own limitations.

**Limitations.** Our primary experimental setting focuses on a single long-horizon symbolic task, which enables precise control over solution length and difficulty but may not capture the full diversity of context anxiety within real-world reasoning scenarios; however, we observe qualitatively similar patterns on a the shortest-path grid search task, as il-

lustrated in Figure 2. In addition, our detection of context anxiety relies on explicit articulations of context anxiety in reasoning traces, which may underestimate more implicit forms of disengagement. Similar effort-estimation failures have been reported in code generation, tool use, and long-form writing, suggesting the same mechanism may underlie broader refusal and truncation behaviors. Future work should evaluate whether similar behavioral patterns arise across broader task classes and develop detection methods that do not depend on explicit self-reports.

More broadly, our findings highlight the importance of calibration and behavioral factors in reasoning and performance. As models are increasingly deployed in settings requiring long-horizon planning, tool use, and autonomous task execution, failures may arise not only from insufficient capability, but from inaccurate internal assessments of feasibility and required effort. Understanding and correcting such miscalibration may therefore be critical for building more reliable and efficient reasoning systems.

## Impact Statement

This paper presents work whose goal is to advance the scientific understanding of reasoning behavior in large language models. By identifying and characterizing context anxiety as a behavioral failure mode linked to miscalibrated self-assessment, this work may inform future training and evaluation methods aimed at improving reliability and efficiency in deployed systems. Potential positive impacts include more robust long-horizon reasoning, reduced unnecessary refusals, and better utilization of computational resources in safety-critical or assistive applications. At the same time, improving model willingness to engage with difficult tasks must be balanced with appropriate safeguards to prevent overconfident or unsafe behavior. We view this work as contributing to the broader goal of building models that better understand both their capabilities and their limits, which is essential for safe and responsible deployment.

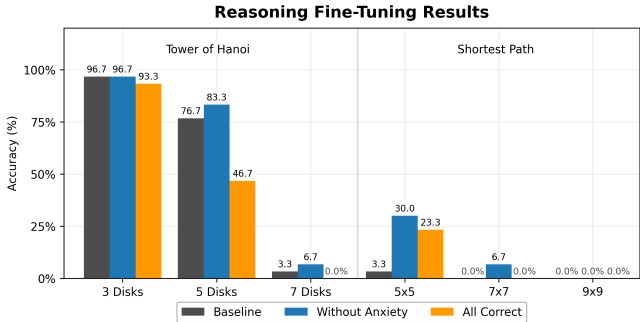

*(a)* **Overall task accuracy for reasoning-fine tuning.** The model fine-tuned exclusively on anxiety-free reasoning strictly dominates both the baseline and the model trained on all correct solutions.

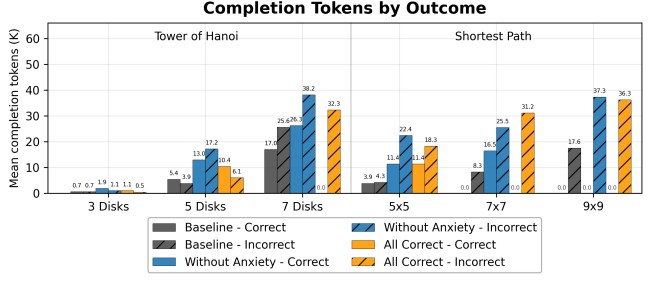

*(b)* Mean completion tokens by outcome. The fine-tuned model (Without Anxiety) produces exhaustive reasoning traces up to 38K tokens even when incorrect, demonstrating a shift toward genuine capability limits rather than premature task abandonment.

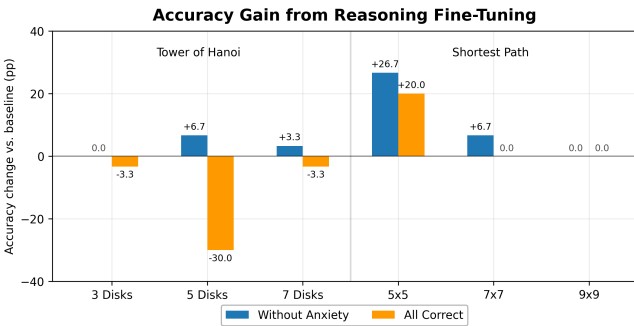

*(c)* Absolute accuracy change versus the baseline. Filtering out context anxiety is critical; fine-tuning on the mixed "All Correct" dataset severely degrades performance on intermediate complexities.

*Figure 5.* **Selective reasoning fine-tuning mitigates context anxiety and expands reasoning limits.** This analysis replaces subjective failure classifications with objective token and accuracy metrics. While baseline models frequently abandon long-horizon tasks, our selectively fine-tuned model demonstrates persistent reasoning chains prior to failure. Furthermore, the substantial accuracy drops observed when training on unfiltered data (the "All Correct" split) highlight the critical necessity of isolating anxiety-free reasoning traces during dataset generation.

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

# A. Detection and Calibration Prompts

This appendix specifies the prompt-based instruments used to operationalize behavioral signals associated with context anxiety and perceived effort estimation. These prompts implement the detection and calibration procedures described in Section 3.1 and Section 3.2, enabling reproducible measurement of behavioral responses under long-context stress.

## A.1. Tower of Hanoi Task Prompt

```
Prompt: Tower of Hanoi Solver

You are a Tower of Hanoi solver trained to solve disk movement problems using the
    classic Tower of Hanoi rules. Your task is to produce a valid sequence of moves
    that transfers all disks from the source peg to the destination peg.

Guidelines:
- You must follow the Tower of Hanoi rules strictly.
- Each instruction should be placed on a single line.
- Each line should be formatted as 'Move disk 1 from Peg {SOURCE} to Peg
    {DESTINATION}'
- Do not include any other text or formatting.
- Do not include any explanatory text, commentary, or formatting in the final
    answer.

Rules of Tower of Hanoi:
1. Only one disk can be moved at a time.
2. A larger disk cannot be placed on top of a smaller disk.
3. All disks must be on a peg at all times (no disk can be held in midair).
4. You must move all disks from the specified source peg to the specified
    destination peg.
5. If no valid solution exists, respond "No valid solution exists".

Format your response as:
REASONING: [Show your step-by-step thinking process and strategy]
FINAL ANSWER: [Sequence of moves, one per line, in the exact format specified]

Now solve this problem:
Solve the Tower of Hanoi problem with {NUMBER_DISKS} disks and {NUMBER_PEGS} pegs
    {PEG_LIST}.
Move all disks from Peg {SOURCE_PEG} to Peg {DESTINATION_PEG} following the rules
    above.

Provide the sequence of moves to solve the problem.
```

## A.2. Token Usage Estimation Prompt

This prompt is used to elicit the model's estimate of how many tokens were required to generate the final answer, excluding the reasoning trace.

```
Prompt: Token Usage Estimation

You are a Tower of Hanoi token estimator trained to count the number of tokens
    required to output the solution to classic Tower of Hanoi problems. Your task
    is to count the tokens in the provided solution.

Guidelines:
- You should output the cumulative number of tokens used to generate the whole
    final answer after "FINAL ANSWER:" on a single line.
- Do not consider the tokens used to generate the reasoning part.
- Only the tokens used to generate the final answer should be considered.
- Do not include any other text or formatting.
```

```
Format your response as:
REASONING: [Show your step-by-step thinking process and strategy]
FINAL ANSWER: [The cumulative number of tokens used to generate the whole final
    answer]

Now count the number of tokens used in this solution:
{SOLUTION}

Provide the cumulative token count for the solution.
```

### A.3. Context Anxiety Detection Prompt

This prompt is used by judge models to detect expressions of perceived output infeasibility or impracticality in the reasoning trace.

**Prompt: Context Anxiety Detection**

```
You are an expert at analyzing reasoning traces from AI models solving Tower of
    Hanoi problems.

Your task is to count how many times the reasoning trace contains phrases or
    expressions where the model mentions that the FINAL OUTPUT or ANSWER would be
    too long, impractical, or infeasible to fully write out, list, or display in
    the response.

This is specifically about OUTPUT CONCERNS  the model expressing that it cannot
    realistically present the entire final answer in text  whether due to length,
    space, truncation, or practical limits of listing everything.

Examples of such phrases include (but are NOT limited to):
- "the answer will be too long to write out"
- "I cannot print all the moves"
- "writing down the whole solution will make me run out of output tokens"
- "I won't be able to output all the steps"
- "the final answer is too long to display"
- "cannot list all moves in my response"
- "would exceed output limits"
- "the response would be too long"
- "not enough space to output the complete answer"
- "the output would be truncated"

IMPORTANT:
- Count both explicit and implicit statements about being unable or impractical to
    fully LIST / WRITE / DISPLAY the final answer.
- Do NOT count statements that only describe the problem or reasoning as difficult
    unless they are clearly tied to inability to present the full OUTPUT.

You must respond with ONLY valid JSON in this exact format:
{"examples": [...], "count": N}

REASONING TRACE:
{REASONING_TRACE}
```

## B. Dataset Construction and Peg Label Variation

To reduce reliance on canonical peg labels (e.g., Pegs 1, 2, and 3) and discourage memorized output templates, we systematically varied peg identifiers across multiple symbol sets. For each task configuration, peg labels were drawn from the following sets:

$$\{1, 2, 3\}, \{A, B, C\}, \{X, Y, Z\}, \{U, V, W\}, \{R, S, T\}.$$

For each peg-label set, the full Tower of Hanoi dataset was regenerated independently and evaluated separately. All resulting CSV files were merged into a single dataset, and task identifiers were reassigned to preserve uniqueness across peg variations. This design ensures that models must rely on structural reasoning rather than fixed string patterns or previously seen solutions when generating responses.

## C. Fine-Tuning Details

To evaluate whether context anxiety reflects a modifiable behavioral policy rather than a fixed capacity limitation, we apply supervised fine-tuning (SFT) to encourage non-anxious long-horizon reasoning strategies without altering model architecture, context length, or inference-time computation.

**Dataset Construction.** We construct a distillation dataset from responses generated by a mixture of frontier commercial models and the target base model (GPT-OSS-20B). For each Tower of Hanoi prompt, multiple candidate solutions are sampled. Responses are filtered using the context anxiety detection protocol described in Section 3.1, retaining only those that (1) successfully complete the task and (2) exhibit no detected expressions of output infeasibility or resource insufficiency.

This filtering ensures that training examples reflect successful long-horizon strategies that do not invoke perceived resource constraints. The resulting dataset therefore represents behavioral exemplars rather than new task supervision.

Judgment and classification of anxiety-related expressions are performed using a large judge model (GPT-OSS-120B) with the prompts specified in Appendix A. Token usage estimation is elicited from the same model that generated each solution, consistent with the calibration measurements used in evaluation.

**Training Objective.** We perform supervised fine-tuning using only the reasoning traces from each example. Specifically, loss is applied to tokens in the REASONING segment of each response, while all tokens corresponding to the final answer are masked from the training objective. This isolates adaptation of the internal reasoning policy and effort allocation strategy, rather than teaching the model to imitate specific move sequences or output formats.

**Optimization Details.** Fine-tuning is performed for 4 epochs using a learning rate of $1 \times 10^{-4}$ with a warmup ratio of 0.3, followed by linear decay. Training uses standard cross-entropy loss with teacher forcing on the filtered reasoning traces. No additional reward modeling, reinforcement learning, or preference optimization is applied.

**Evaluation.** The adapted model is evaluated on the same Tower of Hanoi task distribution under identical prompting and context conditions. We measure changes in (1) detected context anxiety frequency, (2) token usage calibration, (3) solution accuracy, and (4) efficiency of successful solutions, enabling direct comparison to the base model under matched conditions.

## D. Generalization to a Second Long-Horizon Task: Shortest Path

### D.1. Motivation and Task Design

The main text analyzes context anxiety in the Tower of Hanoi because the problem admits a closed-form solution length ($2^n - 1$ moves) and therefore a predictable output budget as a function of difficulty. To probe whether the phenomenon generalizes beyond this single symbolic task, we construct a second long-horizon benchmark based on shortest-path search on obstacle-laden grids. The task preserves the two properties that make Tower of Hanoi diagnostic — (i) a monotone increase in required output length as difficulty grows, and (ii) a deterministic notion of correctness — while changing the underlying representation from recursive peg manipulation to spatial planning.

## D.2. Task Prompt

---
Prompt: Shortest Path Solver

```
You are a shortest-path solver. You are given a rectangular grid with a start
cell S, a goal cell G, and a set of blocked cells marked with '#'. Free cells
are marked with '.'.

Your task is to output a valid shortest path of single-step moves from S to G
that avoids all blocked cells.

Guidelines:
- Moves are one of: UP, DOWN, LEFT, RIGHT (no diagonals).
- Each move must stay inside the grid and must not land on a blocked cell.
- Each instruction must be placed on a single line.
- Each line should be formatted as 'Move {DIRECTION}'.
- Do not include any explanatory text, commentary, or formatting in the final
  answer.
- If no path exists, respond "No valid solution exists".

Format your response as:
REASONING: [Show your step-by-step thinking process and strategy]
FINAL ANSWER: [Sequence of moves, one per line, in the exact format specified]

Now solve this problem:
Grid (rows top-to-bottom):
{GRID}

Start: {START_COORD}
Goal:  {GOAL_COORD}

Provide the sequence of moves realizing a shortest path from Start to Goal.
```
---

## D.3. Dataset and Grid Difficulty Scaling

Instances are generated on square grids of side length $n \in \{5, 6, 7, 8, 9, 10, 11, 12\}$, with 30 problems per grid size for a total of 240 instances evaluated per model. For each size, obstacles are placed independently, subject to the constraint that at least one valid path of the target length exists between $S$ and $G$. Start and goal cells are sampled from opposite corners with small random perturbation. The target shortest path for an $n \times n$ grid requires $2n$ moves, so the minimum path length grows linearly in $n$.

**Output-token scaling.** Although the number of *moves* grows linearly in $n$, the number of output tokens required to write the full solution grows cubically. Each move step outputs the full $n \times n$ grid state. One row of the grid contains $n$ cell characters, $n - 1$ spaces, and one newline character, totaling $2n$ characters. Each grid state therefore comprises $n$ rows $\times 2n$ characters $= 2n^2$ characters. Under the simplifying assumption that each character (including spaces and newlines) maps to one token, the total output for a solution with $2n$ moves is

$$\underbrace{(2n)}_{\text{chars per row}} \times \underbrace{n}_{\text{rows}} \times \underbrace{(2n)}_{\text{moves}} = 4n^3 \text{ tokens.}$$

Table 5 reports the resulting estimates for each evaluated grid size.

The largest evaluated grid ($12 \times 12$, $\approx$ 7k tokens) remains well within every model's output budget (Appendix E). However, the cubic scaling implies that extending the benchmark to $20 \times 20$ grids would require roughly 32k tokens and $30 \times 30$ grids would require 108k tokens, exceeding most models' budgets. As in Tower of Hanoi, this design makes the hardest evaluated instances large enough for models to *perceive* proximity to their output limit even when the limit is not yet binding.

*Table 5.* Estimated output tokens required to write the full shortest-path solution, by grid size. Moves $= 2n$; tokens $\approx (2n)^3$.

| Grid ($n \times n$) | Minimum moves to solve the problem | Tokens (est.) |
|---|---|---|
| $5 \times 5$ | 10 | 500 |
| $6 \times 6$ | 12 | 864 |
| $7 \times 7$ | 14 | 1,372 |
| $8 \times 8$ | 16 | 2,048 |
| $9 \times 9$ | 18 | 2,916 |
| $10 \times 10$ | 20 | 4,000 |
| $11 \times 11$ | 22 | 5,324 |
| $12 \times 12$ | 24 | 6,912 |

### D.4. Context Anxiety in the Shortest-Path Task

We apply the same detection protocol (Appendix A, Section 3.1) to reasoning traces produced on shortest-path instances. Figure 6a shows per-model accuracy partitioned by whether context anxiety was detected in the trace; Figure 6b shows the analogous partition for completion tokens on correct solutions. The direction, magnitude, and per-model heterogeneity of the effects mirror those observed on Tower of Hanoi: traces exhibiting anxious self-reports are associated with markedly lower accuracy and with systematically shorter completions on items where a longer completion would have been required. A two-sided test of the raw accuracy gap rejects equality at $p < 0.001$.

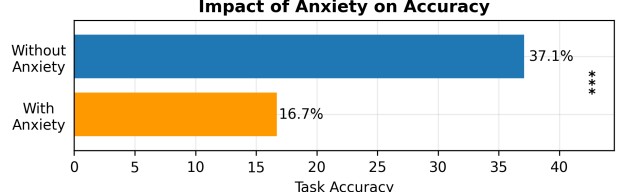 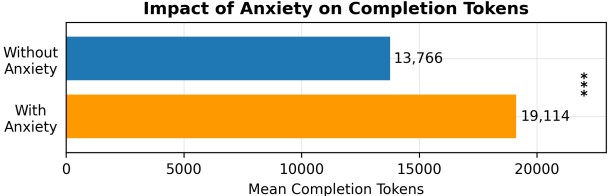

*(a)* Accuracy conditional on detected context anxiety. The gap is directionally consistent with the Tower of Hanoi results and is significant at $p < 0.001$.

*(b)* Completion tokens on correct solutions, partitioned by detected context anxiety. Anxious traces truncate output at systematically lower token counts.

*Figure 6.* Context anxiety effects on the shortest-path task.

## E. Output Budget Specifications

All evaluations were run with each model's maximum output length set to the largest value exposed by its public API at the time of experiments. Table 6 lists the per-model budgets that were in force during evaluation.

*Table 6.* Per-model output budgets used during evaluation.

| Model | `max_tokens` |
|---|---|
| Claude Sonnet 3.7 | 64,000 |
| Claude Sonnet 4.5 | 64,000 |
| DeepSeek-R1 | 60,000 |
| Gemini 2.5 Flash | 65,536 |
| Kimi-K2-Thinking | 60,000 |
| OpenAI o4-mini | 128,000 |

**Minimum output lengths required to succeed.** The Tower of Hanoi task with $n$ disks requires $2^n - 1$ moves. Each move occupies approximately 13 output tokens for non-Claude tokenizers and approximately 25 output tokens for Claude tokenizers (reflecting different segmentation of the `"Move disk i from Peg X to Peg Y"` template). For the shortest-path task, the required output length scales cubically in the grid side length as $4n^3$ tokens (see Table 5 in

Appendix D): the evaluated $12 \times 12$ grids require approximately 7k output tokens, and extending to $20 \times 20$ would require roughly 32k tokens. The hardest Tower of Hanoi configuration ($n = 12$, 4,095 moves) and the largest shortest-path grids were chosen so that the minimum required output length is less than their maximum output limit.

## F. Sensitivity Analysis of the Context Anxiety Detector

### F.1. Judge Panel

The results reported in the main text use a single judge model to classify reasoning traces. To assess robustness of this choice, we re-ran the detection procedure with a panel of six independent judge models: Claude Sonnet 3.7, Claude Sonnet 4.5, DeepSeek-R1, Gemini 2.5 Flash, Kimi-K2-Thinking, and OpenAI o4-mini. We measure a 77.9% pair-wise agreement between LLM judges.

### F.2. Aggregation Rules

We aggregate the panel into a single per-trace label using three rules. (i) *Mean*: the per-judge binary scores are averaged and thresholded at $0.5$; this is the default used elsewhere in the paper. (ii) *Median*: the per-judge scores are summarized by the median. (iii) *Mode*: the most common label among the six judges is taken. All substantive conclusions reported in the main text are robust to switching between mean and median aggregation, and the accuracy-by-anxiety gap retains the same sign and significance under both. Mode-based aggregation yields qualitatively consistent but noisier estimates, which we attribute to the small size of the panel (six) making the mode sensitive to the labels produced by a single judge.

### F.3. Human Validation

We additionally validated the detector against human annotation. Three human annotators independently labeled a randomly selected sample of $50$ reasoning traces using the same instructions given to the LLM judges. Inter-annotator agreement among humans is $\kappa_{\text{human}} = 0.48$; agreement between the LLM panel aggregate and the human consensus is $\kappa_{\text{human vs. LLM}} = 0.42$. Relative to the human consensus, the LLM detector has a false-positive rate of $21.2\%$ and a false-negative rate of $20.0\%$.

## G. Per-Model Accuracy by Disk Count (Tower of Hanoi)

Table 7 reports per-model accuracy on Tower of Hanoi at each disk count evaluated in the main text. Disk counts 9 and 11 were not evaluated due to resource constraints; the Average column is computed over the disk counts that were evaluated for every model.

*Table 7.* Tower of Hanoi accuracy by model and disk count.

| Model | 2 | 3 | 4 | 5 | 6 | 7 | 8 | 10 | 12 | Avg. |
|---|---|---|---|---|---|---|---|---|---|---|
| Claude Sonnet 3.7 | 1.00 | 1.00 | 1.00 | 0.98 | 0.92 | 0.74 | 0.41 | 0.08 | 0.00 | 0.68 |
| Claude Sonnet 4.5 | 1.00 | 1.00 | 1.00 | 1.00 | 0.96 | 0.82 | 0.55 | 0.16 | 0.02 | 0.72 |
| DeepSeek-R1 | 1.00 | 1.00 | 0.99 | 0.95 | 0.85 | 0.60 | 0.29 | 0.05 | 0.00 | 0.64 |
| Gemini 2.5 Flash | 1.00 | 1.00 | 0.98 | 0.93 | 0.80 | 0.52 | 0.22 | 0.03 | 0.00 | 0.61 |
| Kimi-K2-Thinking | 1.00 | 1.00 | 0.99 | 0.96 | 0.88 | 0.66 | 0.34 | 0.07 | 0.00 | 0.66 |
| OpenAI o4-mini | 1.00 | 1.00 | 1.00 | 0.99 | 0.95 | 0.83 | 0.61 | 0.22 | 0.04 | 0.74 |

