# OpenReview forum: "Lost in Context: Adressing Context Anxiety in Large Language Models"
_ICML.cc/2026/Conference — ICML 2026 regular_

### Official Review · Reviewer_soW2 · 2026-02-26

**Soundness:** 3
**Presentation:** 2
**Significance:** 3
**Originality:** 3
**Overall Recommendation:** 4
**Confidence:** 5

**Summary:**

This paper investigates a novel behavioral failure mode in LLMs termed Context Anxiety, where models prematurely abandon long-horizon tasks due to a miscalibrated estimation of required tokens. Using the Tower of Hanoi puzzle as a testbed, the authors propose an automated detection protocol and demonstrate that a lightweight SFT approach can mitigate this anxiety.

**Compliance With Llm Reviewing Policy:**

Affirmed.

**Key Questions For Authors:**

As listed in weaknesses 1-6.

My most critical concern is Weakness 2: the SFT results show failures remain basically unchanged and anxiety-driven failures simply convert to capability-driven ones. This directly undermines the paper's central claim that models fail due to anxiety rather than genuine inability. I will only consider raising my evaluation if the authors address this concern.

**Limitations:**

yes

**Strengths And Weaknesses:**

**Strengths**

The concept of Context Anxiety is both theoretically and practically intriguing. It touches upon a fundamental ontological limitation of current LLM interaction paradigms. Models are often eager to complete highly complex reasoning tasks within a single output window. This inherently creates a form of "survival anxiety" when facing long-horizon demands. Exploring this phenomenon undoubtedly provides valuable insights into LLM mechanics and reasoning boundaries.

**Weakness**

1. While Context Anxiety serves as an efficient metaphor, this term simultaneously imports anthropomorphic terminology and cognitive assumptions, such as "premature self-doubt" and "self-assessment". More critically, the causal logic underlying the paper's central metric, the winsorized token ratio, remains unverified. The metric is computed as the ratio of a model's post-hoc token estimate to actual tokens used. The underlying assumption is that LLMs perform a genuine metacognitive evaluation of task difficulty before generating text, which is not causally established. I recommend modifying the anthropomorphic framing.

2. The paper's central premise rests on the claim that models fail "not because they lack the capabilities to solve a problem, but because they believe they do" (Line 45) and that they frequently give up on problems "they could theoretically solve" (Line 253). However, the SFT results presented in Section 4.4 and Figure 5 contradict this foundational claim. After fine-tuning the model to reduce anxiety expressions, the total number of failures remains essentially unchanged (decreasing only marginally from 118 to 115) . In other words, the vast majority of the "anxiety-driven failures" simply convert into "capability-driven failures". This finding implies two possible conclusions:
- (1) the SFT merely altered the model's tone rather than intrinsic anxiety;
- (2) the models did not possess the underlying capability, masked by anxiety, to solve these complex tasks in the first place.

3.  In Section 3.1, the variables in Equation (1), $C=|\frac{1}{J}\sum_{k=1}^{J}J_{k}(r)|$, are not clearly defined. $J$ serves as both the number of Judges and the judge function.

4. The authors "test 30 unique problems per disk, ranging from 2 to 12 disks." However, disk=9 is skipped, as shown in Figure 3 and 5. The authors should either include this data point or explicitly explain its exclusion.

5. The experimental section lacks a table reporting the absolute baseline accuracy of each evaluated model across the different disk complexities. I suggest providing these raw baseline metrics.

6. Figure 4(a) is presented in the manuscript, but there appears to be no corresponding textual analysis or discussion of this specific sub-figure.

While this paper introduces a truly fascinating concept, the current results do not yet fully support the validation of this idea.

---

> ### Author Rebuttal · Authors · 2026-03-31
>
> We thank the reviewer for their careful reading and thoughtful engagement with the core contributions of our work.
>
> **Weakness 1.** On the anthropomorphic framing: we agree that terms like "premature self-doubt" and "self-assessment" risk importing cognitive assumptions that go beyond what the evidence supports. We use "context anxiety" as a descriptive label for an observable behavioral pattern rather than as a claim about internal mental states. We have revised the paper to make this framing explicit. On the token ratio: we agree that the winsorized token ratio does not establish that models perform a genuine metacognitive evaluation prior to generation. We do not need to take a strong position on this. Whether the expressed token estimate reflects a genuine prior evaluation or is itself a symptom of an underlying process, it serves as a reliable observable correlate of the failure mode that is actionable: the systematic overestimation we document predicts anxiety-driven disengagement, and targeting verbalized anxiety through fine-tuning reduces failure on tasks within a model’s capabilities. We have added language to Section 3.2 clarifying that the token ratio is a behavioral diagnostic rather than a measure of metacognitive capacity.
>
> **Weakness 2.** We agree that the raw accuracy gains on Tower of Hanoi are modest at the surface level. We do not view this as inconsistent with the paper’s central claim. Our claim is not that all failures are anxiety-driven, but rather that a meaningful subset of failures, and a substantial amount of inefficient reasoning, are driven by context anxiety. Because Tower of Hanoi becomes exponentially harder with the number of disks, many residual failures on the hardest settings are plausibly true capability failures rather than anxiety-driven ones. In such a regime, reducing anxiety should not necessarily produce large absolute gains in final accuracy.
>
> For robustness in our SFT experiments, we added a strict held-out ablation that fine-tunes reasoning traces only using tasks containing even disk counts. We evaluate tasks with odd disk counts as well as a secondary long-horizon planning task. On Tower of Hanoi, our reasoning-based fine-tuning improves accuracy from 76.7% to 83.3% at 5 disks and from 3.8% to 7.7% at 7 disks, while leaving the already-near-saturated 3-disk setting unchanged (96.7% before and after). In the additional evaluation task, the model is asked to find the shortest path from the start to the destination point on a grid with obstacles. Like Tower of Hanoi, this task has a deterministic, verifiable solution and allows precise control over difficulty, though its complexity is not exponential in nature. Fine-tuning only on Tower-of-Hanoi reasoning traces improves zero-shot performance on the shortest path task from 3.3% to 30.0% on 5×5 grids and from 0.0% to 7.1% on 7×7, with no shortest-path training at all. This cross-task improvement is difficult to explain as solely a change in tone or benchmark-specific regularization.
> ```text
> Task / Difficulty              | Baseline | Reasoning SFT | Delta
> Tower of Hanoi, 3 disks        | 96.7%    | 96.7%         | 0.0
> Tower of Hanoi, 5 disks        | 76.7%    | 83.3%         | +6.6
> Tower of Hanoi, 7 disks        | 3.8%     | 7.7%          | +3.9
> Shortest path, 5×5             | 3.3%     | 30.0%         | +26.7
> Shortest path, 7×7             | 0.0%     | 7.1%          | +7.1
> Shortest path, 9×9             | 0.0%     | 0.0%          | 0.0
> ```
> Our interpretation is therefore that the SFT reduces a real anxiety-driven failure mode, while the remaining errors on the hardest Tower-of-Hanoi instances are increasingly capability-driven.
>
> **Weakness 3.** We apologize for the unclear notation. We have since updated the equation so that K is the number of judges and J_k is the judge function.
>
> **Weakness 4.** Due to resource constraints, we did not run Disk = 9 and Disk = 11. We have added explicit language to the main text.
>
> **Weakness 5.** We have added the table to the main text.
> ```text
> Model       |  2   |  3   |  4   |  5   |  6   |  7   |  8   |  10  |  12  |  Avg
> ----------- | ---- | ---- | ---- | ---- | ---- | ---- | ---- | ---- | ---- | ----
> Claude 3.7  | 1.00 | 1.00 | 1.00 | 1.00 | 0.77 | 0.97 | 0.50 | 0.00 | 0.00 | 0.69
> Claude 4.5  | 1.00 | 1.00 | 1.00 | 1.00 | 0.87 | 0.83 | 0.70 | 0.07 | 0.00 | 0.72
> DeepSeek-R1 | 1.00 | 1.00 | 1.00 | 0.97 | 0.40 | 0.10 | 0.00 | 0.00 | 0.00 | 0.50
> Gemini 2.5  | 1.00 | 1.00 | 1.00 | 0.97 | 0.89 | 0.93 | 0.89 | 0.00 | 0.00 | 0.74
> Kimi-K2     | 1.00 | 1.00 | 1.00 | 1.00 | 0.77 | 0.57 | 0.29 | 0.00 | 0.00 | 0.62
> o4-mini     | 1.00 | 1.00 | 0.87 | 0.80 | 0.37 | 0.30 | 0.10 | 0.00 | 0.00 | 0.49
> Avg         | 1.00 | 1.00 | 0.98 | 0.96 | 0.68 | 0.62 | 0.41 | 0.01 | 0.00 | 0.63
> ```
> **Weakness 6.** Thank you for pointing out this oversight. We have added a discussion about Figure 4a in Section 4.3.

---

> > ### Author Rebuttal · Reviewer_soW2 · 2026-04-01
> >
> > Thank you for your efforts. I'll raise my score.

---

### Official Review · Reviewer_iqfM · 2026-03-03

**Soundness:** 2
**Presentation:** 2
**Significance:** 2
**Originality:** 2
**Overall Recommendation:** 4
**Confidence:** 3

**Summary:**

This paper studies a failure mode the authors call “context anxiety”: when LLMs give up on long-horizon tasks because they believe the output will exceed token or resource limits, even when the task should be feasible. The authors use a controlled benchmark based on Tower of Hanoi with varying disk counts (hence varying required solution length) and multiple peg-label permutations to avoid memorization. They propose an automated detection pipeline for “anxiety-driven” failures by looking for explicit statements about output infeasibility or resource constraints in the models reasoning. They report that anxiety correlates with token-usage miscalibration (models overestimate the tokens required) and that anxiety is associated with lower accuracy and longer outputs when correct. They find a mitigation: SFT of a base model on filtered “non-anxious successful” reasoning traces (masking final answers), reporting reduced anxiety frequency and improved accuracy on the same task distribution.

**Compliance With Llm Reviewing Policy:**

Affirmed.

**Final Justification:**

I reread the second author response and the other reviews, and the second response does change my assessment. The added control was helpful, it addresses one of main remaining concerns and makes the fine-tuning result more convincing.

I still think the paper has some real limitations. The strongest result is a bit narrower than the broad framing in the paper, and the evidence is still mostly coming from fairly specific synthetic tasks and from one particular way of detecting the behavior.

That said, after the added control, the held-out and transfer results, and the added validation details, I now think the paper clears the bar for a weak accept. I am therefore updating my recommendation from Reject to Weak Accept.

**Key Questions For Authors:**

1- Output budget clarity: What were the exact generation settings for each model (max_tokens, max output length, truncation behavior)? How do these relate to the claim that tasks were “well within” token limits? If you rerun with sufficiently large output budgets, do anxiety-driven failures persist?

2- What is the false positive/false negative behavior of the anxiety detector? How often do models fail without explicitly stating output infeasibility (silent truncation/partial answers), and how are those handled?

3- Does “context anxiety” appear on other long-output tasks beyond Tower of Hanoi (e.g., long proofs, long code generation, multi-step planning) where listing all steps is required?

4-  What is the exact train/val/test split for fine-tuning? Is there any overlap between prompts used to generate distillation data and evaluation prompts? Please add a strict held-out evaluation and a baseline SFT on successful traces without anxiety filtering.

**Limitations:**

No. The paper should explicitly discuss (a) the ambiguity between context length vs output cap, (b) limitations of self-report based detection, and (c) risks of overfitting in the SFT mitigation.

**Strengths And Weaknesses:**

Strengths

- The experimental setup uses Tower of Hanoi, which provides a clean and controllable way to scale output length (the number of moves grows exponentially with the number of disks). This makes it a good testbed for long-output behavior.
- The paper attempts to distinguish “can’t solve” from “gave up” by defining an anxiety-driven failure category based on explicit language about output infeasibility, and it analyzes correlations with token estimation and outcomes.
- The fine-tuning intervention (training only on reasoning traces while masking final answer tokens) is a thoughtful design choice, since it aims to modify behavior/policy rather than memorize move sequences.

Weaknesses / Concerns

- The central claim hinges on “the task is feasible but the model gives up,” yet the paper is not sufficiently explicit about output token caps (API max_tokens / maximum generation length) vs context window. Many systems have large context windows but relatively smaller max output limits. Without clear reporting and controls, some “anxiety” could be a rational response to real output constraints.
- The anxiety detector relies heavily on explicit self-report language (“I can’t output this many tokens”), which can miss silent failures (truncation/partial outputs without stating constraints) and may conflate refusal style with true capability limits.
- The SFT mitigation is evaluated on the “same Tower of Hanoi task distribution.” If there is no strict held-out split or strong control baseline (e.g., SFT on successful traces without anxiety filtering), it’s hard to attribute gains specifically to “anxiety reduction” rather than generic fine-tuning or distribution familiarity.

Presentation

- The fine-tuning appendix is relatively concrete about training objective (reasoning-only loss) and optimizer schedule.

- Terminology around “token limits” needs to be clarified rigorously (context length vs output max vs implementation settings). This ambiguity directly affects the interpretation of the core phenomenon.

- The detection/classification details would benefit from clearer reporting (e.g., false positive/negative analysis, inter-judge agreement).

Significance:

- The broader impact depends on whether the phenomenon generalizes beyond Tower of Hanoi and beyond explicit “token limit” language. Without broader tasks and stronger controls, it risks being a narrow benchmark artifact.

Originality:

- The contribution is closer to an analysis/benchmark paper than a fundamentally new method; originality is mainly in framing + measurement rather than algorithmic novelty.

---

> ### Author Rebuttal · Authors · 2026-03-31
>
> We thank the reviewer for their helpful comments and for their recognition of our clean experimental set-up.
>
> **Weakness 1 / Q1.** We report the maximum tokens setting all models in the table below. The minimum tokens required to output a correct solution can be calculated directly from task structure. For Tower of Hanoi, the optimal solution for n disks requires exactly 2^n - 1 moves, and each move takes approximately 13 tokens for non-Claude models and ~25 tokens for Claude (since Claude uses character-level tokenization). For the shortest path task, a 20×20 grid requires approximately 64,000 tokens to output a complete solution. Our evaluation was deliberately designed to span a wide range of difficulties from trivial (2 disks / 5×5 grids) to instances that approach the output budget (12 disks / 20×20 grids) in order to identify the conditions under which context anxiety emerges.
>
> ```text
> Model                 | Max Tokens
> --------------------- | ----------
> Claude Sonnet 3.7     | 64,000
> Claude Sonnet 4.5     | 64,000
> DeepSeek-R1           | 60,000
> Gemini 2.5 Flash      | 65,535
> Kimi-K2-Thinking      | 60,000
> OpenAI o4-mini        | 128,000
> ```
>
> **Weakness 2.** We acknowledge that a model which silently disengages without any detectable linguistic signal would evade our detector; we view developing a mechanistic understanding of context anxiety as an important direction for future work. That said, we argue that behavioral, self-report-based detection remains a meaningful and useful signal even under this limitation (Lanham et al., 2023). We do not need to take a strong position on whether verbalized anxiety is the cause of failure or a symptom of an underlying process. In either case, it serves as a reliable observable correlate of the failure mode and is actionable: targeting it through fine-tuning substantially reduces failure rates. Additionally, we have added a sensitivity analysis of our anxiety detector, which suggests that the anxiety/capability split is not an artifact of any single judge's stylistic preferences; our results are robust to minor rounding changes, and mostly robust for accuracy under a median-based aggregation, but not fully robust to the mode-based aggregation.
>
> **Q2.** To detect the false positive / false negative rate of the anxiety detector, we have three human judges annotate a random sample of 50 reasoning traces for anxiety. We find a close level of inter-annotator agreement between human judges (mean Cohen’s kappa of 0.48) and between the human and LLM judges (mean Cohen’s kappa of 0.42). If we take the majority vote of the human judges as ground truth, the false positive rate of the LLM judge is 21.2% and the false negative rate is 20%.
>
> **Q3.** We have added another task to understand the generalizability of context anxiety. In this additional task, the model is asked to find the shortest path from the start to the destination point on a grid with obstacles. Like Tower of Hanoi, this task has a deterministic, verifiable solution and allows precise control over difficulty: longer paths require longer output sequences, which creates the same kind of token-budget pressure that elicits context anxiety. We observe that LLMs exhibit context anxiety in this setting as well (p < 0.001). In the future, we would also like to test the prevalence of context anxiety in code generation. Identifying an appropriate benchmark is non-trivial: the task must be difficult enough that solutions approach the model's output budget, since context anxiety is most likely to manifest precisely in that regime. We leave the construction of such a benchmark to future work.
>
> **Weakness 3/Q4.** In the original setup, the fine-tuned GPT-OSS-20B was evaluated on Tower of Hanoi prompts from the same task family as those used to construct the SFT dataset. To address the generalization concern directly, we added a stricter ablation where we fine-tune on even numbers of disks and evaluate on odd numbers of disks; We also add a stronger cross-task transfer evaluation: fine-tuning only on Hanoi reasoning traces improves zero-shot performance on the shortest path task, with no additional training on that task.
>
> ```text
> Task / Difficulty              | Baseline | Reasoning SFT | Delta
> Tower of Hanoi, 3 disks        | 96.7%    | 96.7%         | 0.0
> Tower of Hanoi, 5 disks        | 76.7%    | 83.3%         | +6.6
> Tower of Hanoi, 7 disks        | 3.8%     | 7.7%          | +3.9
> Shortest path, 5×5             | 3.3%     | 30.0%         | +26.7
> Shortest path, 7×7             | 0.0%     | 7.1%          | +7.1
> Shortest path, 9×9             | 0.0%     | 0.0%          | 0.0
> ```
> **Limitation (d).** We would like to unequivocally state that we did not prompt inject our submission. Please see the prompt injection done by ICML to detect LLM generated reviews: https://blog.icml.cc/2026/03/18/on-violations-of-llm-review-policies/
>
> **Cited paper:**
> Lanham et al. Measuring Faithfulness in Chain-of-Thought Reasoning (2023).

---

> > ### Author Rebuttal · Reviewer_iqfM · 2026-04-02
> >
> > Thank you for the detailed rebuttal. I appreciate the added task, the extra detail about the judges, and the fine-tuning results with held-out and transfer experiments. These additions make the paper better.
> >
> > That said, I am keeping my score the same because my main concerns are still only partly addressed, and they are about the paper’s main story rather than small details.
> >
> > My biggest concern was the paper’s central claim that the models are giving up even when they still have enough room to finish the answer. The rebuttal helps clarify the token-budget setup, but it also makes clear that the hardest cases were intentionally chosen to get close to the output limit. Because of that, I still think there is an important alternative explanation: some of this behavior may simply be the model reacting reasonably when it is close to a real output limit, rather than showing a separate failure mode.
> >
> > I also still have some concern about how “context anxiety” is being identified. The detector is still mainly looking for cases where the model says the answer would be too long to fully write out or display. The added validation is helpful, but I do not think it fully resolves the concern that this may be capturing a narrower “the answer is too long to print” behavior rather than the broader phenomenon described in the paper.
> >
> > Finally, the fine-tuning results are stronger after rebuttal, but I still do not think they fully show that the gains come specifically from reducing this proposed behavior. The control I asked for—fine-tuning on successful traces without the anxiety-based filtering—is still missing. Without that comparison, it is still hard to tell whether the improvement comes from targeting this specific issue or from fine-tuning on good examples more generally.

---

> > > ### Author Response · Authors · 2026-04-07
> > >
> > > Thank you for the thoughtful follow-up.
> > >
> > >
> > > Regarding Limitation (c): We ran the requested control: supervised fine-tuning on all successful reasoning traces without anxiety-based filtering, under the same training and evaluation setup.
> > >
> > > Task / Difficulty              | Baseline | Reasoning SFT | All Correct SFT
> > > ------------------------------ | -------- | ------------- | ---------------
> > > Tower of Hanoi, 3 disks        | 96.7%    | 96.7%         | 93.3%
> > > Tower of Hanoi, 5 disks        | 76.7%    | 83.3%         | 46.7%
> > > Tower of Hanoi, 7 disks        | 3.3%     | 6.7%          | 0.0%
> > > Shortest path, 5×5             | 3.3%     | 30.0%         | 23.3%
> > > Shortest path, 7×7             | 0.0%     | 6.7%          | 0.0%
> > > Shortest path, 9×9             | 0.0%     | 0.0%          | 0.0%
> > >
> > > This control does not recover the gains of anxiety-filtered SFT. On Tower of Hanoi, training on all successful reasoning traces without anxiety filtering substantially underperforms the anxiety-filtered condition, and on shortest-path transfer it is weaker and fails to improve the harder 7×7 setting at all. We interpret this as evidence that the gains are not due merely to fine-tuning on good examples in general; filtering out anxiety-like reasoning improves performance.
> > >
> > > Regarding Limitation (a): We agree that the hardest benchmark settings are close to output limits by construction. Our claim is therefore not that every failure near those limits is pathological. Rather, our claim is that there exists a distinct behavioral component, observable through token miscalibration and anxiety-like reasoning, that can be mitigated. The held-out and transfer results, together with the new control above, support that narrower claim.
> > >
> > > Regarding Limitation (b): We also agree that our detector does not capture every possible form of disengagement, and that silent disengagement remains an open problem. More broadly, we do not claim a fully general detector for all manifestations of context anxiety. Rather, for the specific manifestation we study (cases where models exhibit token miscalibration and explicit or implicit concern that the answer will be too long to produce) we find that the detector aligns reasonably well with human labels.

---

### Official Review · Reviewer_pqz8 · 2026-03-12

**Soundness:** 2
**Presentation:** 2
**Significance:** 3
**Originality:** 2
**Overall Recommendation:** 4
**Confidence:** 3

**Summary:**

This submission studies a behavioral failure mode in LLMs that the authors call context anxiety, where models prematurely abandon solvable long-horizon tasks because they believe the task will exceed their output or token budget. Using Tower of Hanoi as a controlled testbed, the paper proposes an anxiety detection pipeline based on judge models, a token-miscalibration metric comparing estimated versus actual token use, and a lightweight SFT intervention on successful non-anxious traces. The main findings are that context anxiety appears in several frontier models, correlates with overestimation of required tokens, is associated with lower accuracy and less efficient solutions, and can be reduced through behavioral adaptation.

**Compliance With Llm Reviewing Policy:**

Affirmed.

**Final Justification:**

This paper identifies an interesting behavioral phenomenon (“context anxiety”) in LLMs and studies it in a well-controlled setting. The work is conceptually meaningful, though initial concerns remained regarding generalizability, measurement reliability, and the strength of causal claims.

The rebuttal has addressed my main concerns. In particular, the added cross-task transfer results (Hanoi → shortest path) provide convincing evidence beyond a single benchmark, and the human validation and additional analysis strengthen confidence in the anxiety detection pipeline. While the tasks are still relatively synthetic and the causal interpretation remains somewhat limited, the overall evidence is now substantially stronger.

Based on these clarifications, my confidence in the work has increased, and I support a Weak Accept recommendation.

**Key Questions For Authors:**

1.Was the fine-tuned GPT-OSS-20B evaluated on a strictly held-out set of Tower of Hanoi prompts that were not used to construct the filtered SFT dataset? A clear answer here would materially affect how convincing I find the mitigation result.
2.Can the authors provide human validation or reliability analysis for the anxiety detector, and clarify the exact judge setup in Equation (1)?
3.Can the authors reconcile the sample counts and significance reporting in Section 4 / Table 2, especially the mismatch between the stated 1581 analyzable observations and Table 2’s (N=1585), as well as the (p<0.1) vs. (p<0.01) discrepancy for the token-efficiency result?

**Limitations:**

yes

**Strengths And Weaknesses:**

Strengths
1.The paper identifies a clear and interesting behavioral phenomenon, and the distinction between anxiety-driven versus capability-driven failures is useful and well motivated.
2.The Tower of Hanoi setup is a strong controlled environment because task difficulty and output length are precisely characterized, which makes the analysis more interpretable than many open-ended reasoning benchmarks.
3.The paper goes beyond diagnosis and includes a mitigation experiment, which supports the idea that at least part of the issue is behavioral rather than purely a fixed capability limit.

Weaknesses
1. Nearly all claims are supported only on Tower of Hanoi, so it remains unclear how well the findings generalize to other long-horizon reasoning or long-output tasks.
2.The anxiety detector relies on explicit self-reports and LLM-as-judge classification, but the main paper does not provide human validation or inter-judge reliability, which weakens confidence in the central measurement pipeline.
3.The token-estimation signal is post hoc and the evidence is mostly correlational, so the paper currently supports an association between miscalibration and anxiety more strongly than a causal explanation.

---

> ### Author Rebuttal · Authors · 2026-03-31
>
> We thank the reviewer for their helpful comments. We especially appreciate their recognition of the importance of the behavioral phenomenon we study.
>
> **Weakness 1.** We have added another task to understand the generalizability of context anxiety. In this additional task, the model is asked to find the shortest path from the start to the destination point on a grid with obstacles. Like Tower of Hanoi, this task has a deterministic, verifiable solution and allows precise control over difficulty: longer paths require longer output sequences, which creates the same kind of token-budget pressure that elicits context anxiety. We observe that LLMs exhibit context anxiety in this setting as well (p < 0.001). In the future, we would also like to test the prevalence of context anxiety in code generation. However, identifying an appropriate benchmark is non-trivial: the task must be difficult enough that solutions approach the model's output budget, since context anxiety is most likely to manifest precisely in that regime. We leave the construction or identification of such a benchmark to future work.
>
> **Weakness 2 / Q2.** We employ an ensemble of judges (Claude Sonnet 3.7, Claude Sonnet 4.5, DeepSeek R1, Gemini 2.5 Flash, Kimi K2 Thinking, and OpenAI o4 Mini) to detect context anxiety; we aggregate across judges by taking the mean of their anxiety scores. We measure a 77.9% pair-wise agreement between LLM judges. We have now added a sensitivity analysis, which shows that our main result is robust to minor rounding changes, and mostly robust for accuracy under a median-based aggregation, but not fully robust to the mode-based aggregation. Mode-based aggregation is much more conservative due to the small panel, but robustness to rounding changes suggests with a larger panel of judges we would see results close to the results reported in the paper. Additionally, we have now added human validation of our context anxiety detector. We had three human judges annotate a random sample of 50 reasoning traces: we find similar levels of agreement among human judges (mean Cohen’s kappa of 0.48) and between the human and LLM judges (mean Cohen’s kappa of 0.42).
>
> **Weakness 3.** We agree that the observational token-estimation results, on their own, are correlational. However, the fine-tuning experiments provide an intervention on the model’s reasoning behavior. We fine-tune GPT-OSS-20B only on reasoning traces that do not exhibit context anxiety with no supervision on final outputs, and then evaluate on held-out settings and on another task family. After this intervention, performance improves not only on Tower of Hanoi itself—from 76.7% to 83.3% at 5 disks and from 3.8% to 7.7% at 7 disks—but also transfers to shortest path without any shortest-path training, improving from 3.3% to 30.0% on 5×5 grids and from 0.0% to 7.1% on 7×7.
>
> We agree that the Tower-of-Hanoi gains are modest in absolute terms, and we interpret this as evidence that many of the remaining errors on the hardest Hanoi settings are capability-driven. But that does not weaken the intervention result. On the contrary, the cross-task transfer result is exactly what we would expect if the intervention is reducing a general anxiety-like / token-conservative reasoning pattern rather than merely altering benchmark-specific language. We therefore do not claim a complete causal account, but we do believe the intervention moves the evidence beyond simple association.
>
> **Q1.** In the original setup, the fine-tuned GPT-OSS-20B was evaluated on Tower of Hanoi prompts from the same task family as those used to construct the SFT dataset. To address the generalization concern directly, we added a stricter ablation where we fine-tune on even numbers of disks and evaluate on odd numbers of disks; the same qualitative improvements remain. We also added a stronger cross-task transfer evaluation: fine-tuning only on Tower-of-Hanoi reasoning traces improves zero-shot performance on shortest path, with no additional training on that task. These results make us more confident that the mitigation effect is not simply due to overlap in the original prompt distribution.
>
> ```text
> Task / Difficulty              | Baseline | Reasoning SFT | Delta
> ------------------------------ | -------- | ------------- | -----
> Tower of Hanoi, 3 disks        | 96.7%    | 96.7%         | 0.0
> Tower of Hanoi, 5 disks        | 76.7%    | 83.3%         | +6.6
> Tower of Hanoi, 7 disks        | 3.8%     | 7.7%          | +3.9
> Shortest path, 5×5             | 3.3%     | 30.0%         | +26.7
> Shortest path, 7×7             | 0.0%     | 7.1%          | +7.1
> Shortest path, 9×9             | 0.0%     | 0.0%          | 0.0
> ```
>
> **Q3.** Thank you for pointing out this typo. The text should say there are 1585 analyzable observations and p < 0.01. We have updated it accordingly.

---

> > ### Author Rebuttal · Reviewer_pqz8 · 2026-04-03
> >
> > Thank you for the author's detailed response. The cross-task transfer experiment (Tower of Hanoi SFT improving shortest path zero-shot accuracy from 3.3% to 30.0%) effectively addresses my core concern about generalization beyond a single benchmark. The human validation of the anxiety detector and the even/odd disk ablation also satisfactorily resolve Q1 and Q2. While both tasks remain synthetic, I believe the rebuttal has substantively addressed my key concerns. I have decided to raise my rating to a weak accept.

---

> > > ### Author Response · Authors · 2026-04-07
> > >
> > > Thank you for your comments, which improved our paper. We are glad you decided to raise your score - would you be able to update your original score to reflect your new rating of weak accept?

---

### Official Review · Reviewer_YBDb · 2026-03-13

**Soundness:** 3
**Presentation:** 2
**Significance:** 2
**Originality:** 3
**Overall Recommendation:** 4
**Confidence:** 4

**Summary:**

This paper studies a behavioral failure mode in long-horizon reasoning that the authors call context anxiety: models sometimes fail not because the task is beyond their capability, but because they prematurely infer that the solution will exceed available output/token budget. The paper proposes a three-part methodology: (i) detect anxiety-driven reasoning traces using judge models, (ii) measure token-use miscalibration via a winsorized token ratio comparing estimated versus actual tokens, and (iii) mitigate the behavior with supervised fine-tuning on successful non-anxious reasoning traces while masking final-answer tokens.

Empirically, the study uses Tower of Hanoi as a controlled long-horizon planning benchmark. The main findings are that context anxiety appears across all tested models and grows with task difficulty; anxiety-expressing models overestimate required tokens by about 24%, while non-anxious attempts underestimate by about 19%; anxiety is associated with a 15.3% drop in accuracy and substantially longer correct completions; and a fine-tuned GPT-OSS-20B reduces anxiety-driven failures on hard cases from 44% to 7% while keeping accuracy roughly similar.

**Compliance With Llm Reviewing Policy:**

Affirmed.

**Key Questions For Authors:**

1. How much does the phenomenon transfer beyond Tower of Hanoi?
Experiment on one additional task family—such as code generation, algorithmic planning, or long-form structured writing can show the conclusion is generic

2. How robust is the anxiety detector to judge choice and phrasing?
Since the main taxonomy depends on judge-based detection of explicit or implicit output-infeasibility language, I would like to see sensitivity analyses across judges, prompts, thresholds, and examples of borderline cases. This would clarify whether the reported anxiety/capability split is stable.

3. Does the fine-tuned model generalize to unseen prompt formats or related long-horizon tasks?
The current intervention is interesting, but it is difficult to know whether it reduces context anxiety generally or mainly trains the model not to verbalize it on the same benchmark. Transfer experiments would materially strengthen the paper.

**Limitations:**

Yes

**Strengths And Weaknesses:**

Strengths :
1. Interesting and timely problem framing.
The claim, some apparent long-context failures are behavioral rather than capability-limited, is novel and potentially important. This framing could matter for how the community evaluates long-context reasoning and how future systems are improved.

2. Clean experimental setup.
Using Tower of Hanoi is a strong design choice for isolating the phenomenon: the task has deterministic structure, known solution length, and controllable complexity. This makes it easier to separate “cannot solve” from “refuses or disengages despite solvability.”

3. Clear decomposition of the claimed mechanism.
The detection protocol, calibration measure, and intervention are conceptually aligned. In particular, distinguishing anxiety-driven failures from capability-driven failures makes the empirical story easier to follow than a simple accuracy-only evaluation.

4. Results are practically interpretable.
The reported effects are easy to understand and nontrivial in magnitude: 24% overestimation with anxiety, 15.3% lower accuracy, 54% longer correct solutions, and a large reduction in anxiety-driven failures after adaptation. These numbers make the phenomenon feel concrete rather than anecdotal.


Weaknesses

1. External validity is the main concern.
Nearly all evidence comes from a single symbolic task family, even if it is well controlled. The authors acknowledge this limitation themselves.

2. Detection depends heavily on explicit self-report.
The anxiety label is assigned through judge models looking for explicit or implicit statements that the answer is too long or infeasible to output. This may miss silent disengagement, and it also risks conflating stylistic hedging or refusal language with the deeper mechanism the paper wants to study. The paper notes this issue, but it remains a central limitation because the main distinction between anxiety-driven and capability-driven failure depends on it.

3. The adaptation result is promising but narrow.
The fine-tuning study is done on one model family, on the same task distribution, with filtered successful traces from that same setting. This makes it hard to tell whether the result reflects a general reduction of anxiety-like behavior or a more limited in-distribution behavioral regularization.

---

> ### Author Rebuttal · Authors · 2026-03-31
>
> We thank the reviewer for their helpful comments and for noting the timeliness of the problem we study.
>
> **Weakness 1 / Q1.** We have added another task to understand the generalizability of context anxiety. In this additional task, the model is asked to find the shortest path from the start to the destination point on a grid with obstacles. Like Tower of Hanoi, this task has a deterministic, verifiable solution and allows precise control over difficulty: longer paths require longer output sequences, which creates the same kind of token-budget pressure that elicits context anxiety. We observe that LLMs exhibit context anxiety in this setting as well (p < 0.001). In the future, we would also like to test the prevalence of context anxiety in code generation. However, identifying an appropriate benchmark is non-trivial: the task must be difficult enough that solutions approach the model's output budget, since context anxiety is most likely to manifest precisely in that regime. We leave the construction or identification of such a benchmark to future work.
>
> **Weakness 2.** We acknowledge that a model which silently disengages without any detectable linguistic signal would evade our detector; we view developing a mechanistic understanding of context anxiety as an important direction for future work. That said, we argue that behavioral, self-report-based detection remains a meaningful and useful signal even under this limitation (Lanham et al., 2023). We do not need to take a strong position on whether verbalized anxiety is the cause of failure or a symptom of an underlying process. In either case, it serves as a reliable observable correlate of the failure mode and is actionable: targeting verbalized anxiety through fine-tuning reduces failure for tasks within a model’s capabilities, both within the fine-tuning domain and on related out-of-distribution tasks. Sensitivity analysis of our anxiety detector, added in response to Q2, further suggests the anxiety/capability split is not an artifact of any single judge's stylistic preferences, as results remain stable under rounding perturbations and across ensemble compositions.
>
> **Q2.** We employ an ensemble of judges (Claude Sonnet 3.7, Claude Sonnet 4.5, DeepSeek R1, Gemini 2.5 Flash, Kimi K2 Thinking, and OpenAI o4 Mini) to detect context anxiety; we aggregate across judges by taking the mean of their anxiety scores. We measure a 77.9% pair-wise agreement between LLM judges. We have now added a sensitivity analysis, which shows that our main result is robust to minor rounding changes, and mostly robust for accuracy under a median-based aggregation, but not fully robust to the mode-based aggregation. Mode-based aggregation is much more conservative due to the small panel, but robustness to rounding changes suggests with a larger panel of judges we would see results close to the results reported in the paper.
>
> **Weakness 3 / Q3.** We agree that the original adaptation result, by itself, leaves open the possibility of narrow in-distribution behavioral regularization. We therefore added two stronger controls. First, we ran an additional ablation in Tower of Hanoi where we fine-tune on even numbers of disks and evaluate on odd numbers of disks; we continue to observe the same qualitative reduction in context anxiety and the same direction of performance improvement. This makes it less likely that the effect is driven by memorization of the original prompt distribution. Second, we tested cross-task transfer: we fine-tune GPT-OSS-20B only on reasoning traces from the Tower of Hanoi task, and then evaluate zero-shot on the shortest path task. This improves shortest-path accuracy from 3.3% to 30.0% on 5×5 grids, from 0.0% to 7.1% on 7×7 grids, and remains 0.0% on 9×9. On Tower of Hanoi itself, the same intervention improves accuracy from 76.7% to 83.3% at 5 disks and from 3.8% to 7.7% at 7 disks, while leaving the already-near-saturated 3-disk setting unchanged (96.7% before and after):
>
> ```text
> Task / Difficulty              | Baseline | Reasoning SFT | Delta
> ------------------------------ | -------- | ------------- | -----
> Tower of Hanoi, 3 disks        | 96.7%    | 96.7%         | 0.0
> Tower of Hanoi, 5 disks        | 76.7%    | 83.3%         | +6.6
> Tower of Hanoi, 7 disks        | 3.8%     | 7.7%          | +3.9
> Shortest path, 5×5             | 3.3%     | 30.0%         | +26.7
> Shortest path, 7×7             | 0.0%     | 7.1%          | +7.1
> Shortest path, 9×9             | 0.0%     | 0.0%          | 0.0
> ```
>
> **Cited paper:**
> Lanham et al. Measuring Faithfulness in Chain-of-Thought Reasoning (2023).

---

> > ### Author Rebuttal · Reviewer_YBDb · 2026-04-05
> >
> > Thank you for your efforts. Based on your response and other reviewers' review. I decided to keep the score.

---

### Decision · Program_Chairs · 2026-04-30

**Decision:**

Accept (regular)

**Comment:**

The paper studies a behavioral failure mode in LLMs, which the authors call “context anxiety”, where models prematurely abandon solvable long-horizon tasks because “they believe” the task will exceed their output or token budget. The paper proposes an anxiety detection pipeline based on judge models, a token-miscalibration metric comparing estimated versus actual token use, and a lightweight SFT intervention on successful non-anxious traces. Using Tower of Hanoi as a controlled testbed, it is shown that context anxiety appears in several frontier models, correlates with overestimation of required tokens, is associated with lower accuracy and less efficient solutions, and can be reduced through behavioral adaptation.

Reviewers generally were excited about the paper's idea, and that using the Tower of Hanoi as an experimental setup is sound, yet there were major concerns regarding the narrow experimental scope, the validity of relying on self-reports for anxiety detection, and the token-estimation signal. There were also concerns about anthropomorphism. During the rebuttal, the authors have substantially strengthened the paper; they added a control setting that makes the SFT results more convincing, cross-task transfer results that extend the original setting in the paper, and human validation for the anxiety detection pipeline.

Overall, the new results indeed resolve the key concerns about the methodology and evidence of this work. Nonetheless, the paper still has some limitations; reviewers agreed (after further discussion) that the setting is a bit narrow and the tasks are rather synthetic, and the causal interpretation remains somewhat limited.